# Dissecting the Molecular Mechanisms Driving Electropathology in Atrial Fibrillation: Deployment of RNA Sequencing and Transcriptomic Analyses

**DOI:** 10.3390/cells12182242

**Published:** 2023-09-09

**Authors:** Fabries G. Huiskes, Esther E. Creemers, Bianca J. J. M. Brundel

**Affiliations:** 1Department of Physiology, Amsterdam UMC, Location Vrije Universiteit, VUmc, Amsterdam Cardiovascular Sciences, Heart Failure and Arrhythmias, 1081 HZ, Amsterdam, The Netherlands; f.g.huiskes@amsterdamumc.nl; 2Department of Experimental Cardiology, Amsterdam UMC, Location AMC, Amsterdam Cardiovascular Sciences, Heart Failure and Arrhythmias, 1105 AZ Amsterdam, The Netherlands; e.e.creemers@amsterdamumc.nl

**Keywords:** atrial fibrillation, RNA sequencing, transcriptomics, pathophysiology

## Abstract

Despite many efforts to treat atrial fibrillation (AF), the most common progressive and age-related cardiac tachyarrhythmia in the Western world, the efficacy is still suboptimal. A plausible reason for this is that current treatments are not directed at underlying molecular root causes that drive electrical conduction disorders and AF (i.e., electropathology). Insights into AF-induced transcriptomic alterations may aid in a deeper understanding of electropathology. Specifically, RNA sequencing (RNA-seq) facilitates transcriptomic analyses and discovery of differences in gene expression profiles between patient groups. In the last decade, various RNA-seq studies have been conducted in atrial tissue samples of patients with AF versus controls in sinus rhythm. Identified differentially expressed molecular pathways so far include pathways related to mechanotransduction, ECM remodeling, ion channel signaling, and structural tissue organization through developmental and inflammatory signaling pathways. In this review, we provide an overview of the available human AF RNA-seq studies and highlight the molecular pathways identified. Additionally, a comparison is made between human RNA-seq findings with findings from experimental AF model systems and we discuss contrasting findings. Finally, we elaborate on new exciting RNA-seq approaches, including single-nucleotide variants, spatial transcriptomics and profiling of different populations of total RNA, small RNA and long non-coding RNA.

## 1. Introduction

At present, selection of an optimal strategy for effective management of atrial fibrillation (AF), the most common age-related cardiac arrhythmia, is challenging [1]. The main reason for AF management failure is that the exact root causes for AF are insufficiently understood and, therefore, effective AF diagnostic instruments and therapies are lacking. As the life expectancy of the worldwide population is increasing, the steep rise in incidence of AF in the general population is becoming an urgent public health issue, especially in the Western world and in parts of the Eastern world. Eventually, AF may severely affect patient’s quality of life, as it is associated with serious complications such as stroke, heart failure (HF), cognitive impairment, and sudden cardiac arrest, which results in increased morbidity and mortality [1].

To improve AF therapy and diagnostics, research is increasingly focused on dissection of the molecular and electrical root causes of AF. Evidence from experimental and clinical AF studies reveal a key role for so-called ‘electropathology’ as a driver of AF. Electropathology is defined as electrical conduction disorders, and consequently contractile dysfunction, that are caused by molecular changes in the atrial tissue that drive structural changes (including myolysis, dilatation and fibrosis) and AF initiation and perpetuation [1]. So far, experimental AF studies identified several key pathways for molecular changes, including derailment in protein homeostasis (proteostasis), stress signaling and inflammasome activation, that result in impairment of cardiomyocyte calcium handling, complex patterns of electrical activation and hence atrial cardiomyocyte contractile dysfunction [2,3,4].

Transcriptome analysis is a powerful tool for identifying molecular disease mechanisms. Extensive AF-related gene expression data have been collected through atrial tissue and peripheral blood microarray studies. Microarray studies on AF and the insights they provide have been comprehensively reviewed by Steenman [5]. During the last decade, microarray studies have gradually been superseded by nucleotide sequencing technologies for transcriptome analysis due to their improved specificity to detect transcripts, and specifically isoforms.

RNA sequencing (RNA-seq) has become a valuable tool for transcriptome-wide analysis of differential gene expression [6,7]. RNA-seq approaches are utilized to study different aspects of RNA biology, including alternative splicing, translation (using ribosome profiling), allelic imbalance, pathway analysis and RNA structure [8]. As such, RNA-seq has contributed to an improved understanding of the molecular mechanisms driving cardiac diseases, including AF. In this review, we provide an overview of RNA-seq studies in the setting of AF, that have been conducted in atrial tissue samples from humans and experimental models. We discuss novel insights retrieved from these studies and we touch upon recent developments in next generation sequencing, such as single-cell RNA sequencing and spatial transcriptomics, which have the potential to significantly transform our understanding of the electropathology of AF.

## 2. RNA Sequencing Studies in Human AF Cohorts

Over the past decade, several RNA-seq studies have been conducted in right and left atrial tissue samples of patients with AF, controls in sinus rhythm (SR) without reported history of AF and patients in SR who developed post-operative AF (PoAF) after open heart surgery. The AF classification does not indicate rhythm at time of surgery. Differences in gene expression are studied to elucidate molecular pathways that drive AF. Table 1 provides an overview of 17 reported RNA-seq studies on human atrial tissue samples comparing AF with SR patients. Details of the tissue samples, AF classifications and underlying heart conditions are also included as these are essential to interpret the results and identify AF specific alterations.

Of all the 17 RNA-seq studies presented in Table 1, only 4 studies make a comparison between the various AF classes, including paroxysmal (PAF) or persistent (PeAF) AF. Furthermore, underlying heart conditions vary between the mentioned studies, which hampers direct comparison in RNA-seq data between the various studies. Additionally, in several studies small group sizes are used in combination with multiple underlying heart conditions, which impedes separation of AF-related expression patterns from those related to the various underlying heart conditions. Separation of AF-related expression patterns is further complicated in studies that have not matched their SR and AF groups, where differential gene expression may reflect underlying cardiac disease over AF. Other factors, such as aging, will provide additional background signal when healthy donor tissue is used as the control.

### 2.1. Large Cohort Studies

The study by Zeemering et al. sequenced the largest collection of atrial tissue samples from PAF (*n* = 53), PeAF (*n* = 51), and SR patients (*n* = 91) [9]. A large proportion of these patients suffered from end-stage HF (40% in SR, 26% in PAF, 49% in PeAF). This led the authors to stratify the cohort according to HF status. Expression profiling of the HF negative stratum revealed 222 differentially expressed genes (DEGs) between SR and PeAF patients and 18 DEGs between PAF and PeAF patients without HF. In HF patients, 125 DEGs were found between the SR and PeAF patients. No DEGs were found when comparing SR and PAF patients in either the negative HF or positive HF stratum. 35 DEGs were found between SR and PeAF patients, independent of HF status, and these genes are considered robust AF-related markers. Interestingly, this study also found that gene expression differences are consistent between LAA and RAA, which is in contrast to the study by Thomas et al. and Hsu et al. who found different AF expression profiles between LAA and RAA [11,12]. Ke et al. re-analyzed the dataset of Thomas and contrarily concluded that in both atria, AF is manifested through the same mechanisms, as the DEGs specific to LAA and RAA are involved in similar pathways [13]. Pathway analysis of the 35 robust, AF-related DEGs found in the study by Zeemering et al. identified genes involved in cardiomyocyte structure, cardiac conduction, fibrosis, inflammation and endothelial dysfunction. As these DEG-related pathways are only found in PeAF patients compared to SR controls, the authors conclude that most transcriptional changes observed in PeAF are a consequence of AF, rather than a cause.

Van den Berg et al. also performed a large sample size RNA-seq study to investigate structural remodeling processes in AF [10]. LAA tissue samples of 22 SR patients undergoing coronary artery bypass grafting (CABG) and/or valve surgery, and of 22 PAF and 20 PeAF patients undergoing thoracoscopic ablation were included. Underlying CABG or valve disease is not reported for the AF groups. Of the 17,324 mRNAs from protein-coding genes that were detected in the atrial tissue samples, 4581 were found to be differentially expressed between PeAF and SR, 1947 between PAF and SR, and 814 between PAF and PeAF. 1423 DEGs were commonly dysregulated in PAF and PeAF when compared to SR. These DEGs may represent genes that are involved in the progression of AF. 41 of the 1423 DEGs were also dysregulated during the progression of AF (between PAF and PeAF). Enrichment analysis of these 41 DEGs revealed the prevalence of downregulated epicardium-associated genes, including epicardial cell markers and epithelial cell–cell junction genes. The authors relate this downregulation of epicardial genes to endothelial-to-mesenchymal transition (EMT). Pathway analysis of DEGs found in both PAF and PeAF to SR comparisons reveals upregulation of angiogenesis and enhanced perivascular gene signatures. Additionally, biosynthesis of extracellular matrix (ECM) components and ECM degrading constituents were both found to be upregulated for AF. Enrichment mapping indicates a central role of integrin signaling in a very complex network that connects EMT, angiogenesis and ECM remodeling through a myriad of signaling pathways. The occurrence of EMT, angiogenesis and ECM remodeling were subsequently confirmed through histological studies, which provided additional spatial context for these processes. The authors concluded that structural remodeling in AF involves a mesenchymal cell response that arises from the epicardium and perivascular niche through processes of EMT.

### 2.2. Differentially Expressed Protein Coding Genes

Other RNA-seq studies found a broad spectrum of AF-related pathways by comparing expression profiles between SR patients and PeAF patients. Enrichment of downregulated genes was found in pathways related to calcium signaling, MAPK signaling and Wnt signaling [11,13,14,15,16,22,25]. Calcium signaling and MAPK signaling have previously been connected with clinical AF in electrical and structural remodeling [26,27,28,29,30]. Upregulated genes were enriched in focal adhesion, MAPK signaling, RAP1 signaling, PI3KT signaling and cardiac muscle contraction pathways. The first pathway is associated with AF-related fibrosis and the latter three are connected to electrical remodeling in AF [31,32,33,34]. Chen et al. and Jiang et al. compared gene expression levels of RA tissue from SR patients with both the RA and LA tissues from PeAF patients [15,16]. They highlight involvement of the PPAR pathway and chloride intracellular channels, respectively, and attribute their role in AF to lipid metabolism and mitochondrial function. Lipovsky et al. compared expression levels in isolated atrial cardiomyocytes between HF patients with and without a history AF, that underwent a heart transplant [23]. They identified LA specific dysregulation of ion channels, driven by Notch activation in AF. Notch signaling is rarely mentioned alongside AF, but it is known to play a role in cardiac development and in chamber-specific regulation of ion channels in the ventricular myocardium [35,36].

### 2.3. Non-Coding RNAs

Besides uncovering the expression profiles of protein-coding genes, RNA-seq has also unlocked unprecedented access to the non-coding part of the transcriptome [37]. This is of significance in AF research, in which genome-wide association studies (GWAS) have pinpointed the majority of genetic AF factors to non-coding regions [38]. Specific interest in non-coding RNAs (ncRNA) and their role in AF progression is based on their potential as gene expression regulators [39]. Functionally important types of ncRNAs include long non-coding RNAs, circular RNAs and microRNAs. It must be noted that functional characterization of ncRNAs in AF is rather limited at the moment, and that evidence from the discussed papers is based on predictions and correlations, and often deduced from adjacency to protein coding genes in the genome. Distinct lncRNA expression profiles for PeAF were found in LAA and RA tissue samples [13,22]. High correlations in expression patterns were found for lncRNA RP11-99E15.2 with ITGB3 and lncRNA RP3-523K23.2 with HSF2, which are involved in ECM-binding and intracellular stress signaling, respectively [40,41]. Integrins such as ITGB3 are implicated in structural remodeling in AF and the heat shock response is intrinsically cardioprotective [42,43,44,45]. Differences in lncRNA expression were also found between PAF and PeAF patients and healthy donor LAA tissues [14]. Pathway enrichment analysis for lncRNA-adjacent mRNAs specifies a combination of structural and electrical remodeling candidate pathways, including Ras/MAPK signaling, arrhythmogenic RV cardiomyopathy, PPAR signaling and signaling pathways for pluripotency of stem cells as dysregulated in AF.

CircRNAs represent another class of non-coding RNAs of which the function is largely unknown [46]. Due to their circular structure circRNAs are protected from degradation by exonucleases and remain more stable than linear RNA molecules. Functionally, it has been shown they can act as miRNA sponges, protein-scaffolding molecules and transcription regulators. Distinct circRNA expression profiles were found between SR and PeAF group for LAA and RAA samples [17,18]. Pathway enrichment analysis for circRNA host genes suggest a role for inflammation, intercellular communication and cellular metabolism, as well as regulation of muscle cell contraction, regulation of calcium signaling, protein modification and cardiovascular tissue development in PeAF [18,19]. Interaction mapping of differentially expressed calcium handling-associated circRNAs with miRNAs predicts multiple interactions for hsa-miR-208 and hsa-miR-23 [18]. The functional association for these two miRNAs to AF stems from their reported involvement in sarcomere and calcium remodeling, and fibrosis, respectively [47,48]. Interaction mapping between differentially expressed miRNAs and mRNAs in the RA of PeAF patients suggests suppression of SDC1 expression by miR302b-3p [21]. This negative interaction was validated in vitro and is reportedly involved in cardiac fibrosis through TGF-β/Smad2 signaling [49,50]. When comparing general circRNA and miRNA expression profiles between PeAF and PAF patients, an interesting trend appears [20]. General miRNA expression is increased in PAF patients and circRNA expression is increased in PeAF, the latter is believed to suppress the availability of miRNAs through sponging. This may suggest that an escalation in epigenetic regulation mechanisms limits gene silencing and is constructive in the progression of AF to a permanent form.

### 2.4. Combining Genetic Variation and Differential Gene Expression Analysis

Hsu and Sigurdsson studied the role of genetic variation on gene expression changes in AF [24,25]. Hsu performed RNA-seq on LAA tissue from a large group of 265 subjects and combined the expression profiles with SNP data [24]. The resulting expression quantitative locus (eQTL) maps enable linking genetic variation with AF-related gene expression patterns. The authors associated genetic variation in loci 10q22.2 and 5q31.2 to calcineurin signaling, sarcomere association and Wnt RhoGAP signaling in AF. Sigurdsson specified differential gene expression in a study on mechanisms involved in developing post-operative AF (PoAF) [25]. Differential expression of 23 genes was found in comparison between patients who did or did not develop PoAF. DEGs could be separated into two functional classes: the Wnt pathway and GMP metabolism.

In conclusion, based on the differentially expressed gene-related pathways, several molecular mechanisms that manifest in the atria at cellular and tissue level are associated with AF occurrence. Downregulated pathways recurring between studies are mostly signaling pathways that guide a cellular response to an extracellular signal, including MAPK signaling and Wnt signaling. Upregulated pathways include intracellular signaling pathways (PI3KT-akt, Rap1), cellular response to external stimuli (MAPK), and cardiac muscle contraction. The challenge remains to differentiate between factors in the onset and progression of AF with collateral effects of AF. Most AF-related pathways identified in RNA-seq studies are involved in tissue repair. In depth evaluation of differential gene expression in the PAF group and comparison with PeAF patients may help to separate cause and effect. Applying this AF progression stratification to (e)QTL studies or in cross-referencing transcriptomics results with GWAS studies may provide another opportunity to identify potentially causal mechanisms for AF onset and progression. Molecules in the Wnt signaling pathway are interesting candidates, as they are associated with AF through differential expression and genetic variation and they seem related to development of PoAF as well [11,20,25]. Experimental data on the involvement of Wnt signaling in AF indicates dysregulation of Wnt5 and B-catenin in atrial adipositas and atrial fibrosis, respectively [51,52,53].

## 3. Transcriptomic Analyses in Experimental AF Model Systems

RNA-seq approaches have also been used in experimental animal and cell model systems to gain insights in the molecular mechanisms of AF, as summarized in Table 2. Comparing the outcomes of AF model studies to AF patient cohort studies may shine a light on the molecular root causes of AF. In addition, this comparison may reveal potential blind-spots of patient cohort RNA-seq studies, as patient cohorts can be highly variable in disease severity, progression, age, gender and medication use. These factors are controlled in animal studies, offering the advantage to study subtle cellular processes or pathways. For an overview of the animal models utilized in AF research, we refer to a recent review on this topic by Schüttler et al. [54].

### 3.1. Scope of Cell, Rodent and Large Animal Model Research

In general, the AF model systems can be assigned into three groups: human iPSC atrial CMs, rodent models and large animal models. The types of RNA-seq studies performed on these models are either a direct comparison between induced or simulated AF and controls, or studies on the effect of a treatment or a specific mutation. Differential gene expression studies in AF models in rats and rabbit suggest involvement of cell cycle-related processes, inflammatory signaling, phagosome and calcium signaling pathways [58,62]. Canine and sheep studies have captured a rich collection of potential AF-related transcripts, including pathway elements for muscle structural development, ECM structure, intercellular glutamate and SLIT/ROBO signaling, intracellular calcineurin signaling and ion channels [64,65].

### 3.2. Treatment and Mutation-Related Pathways

Treatment of mouse AF models with curcumin or colchicine, which are considered to protect against AF, affect gene expression in pathways associated with inflammatory regulation and fibrosis, and IL-17 signaling and renin secretion pathways, respectively [59,60]. The findings indicate a possible role for these pathways in experimental AF. More relevant is the differential expression that was found in inflammatory signaling, fibrosis, lipid metabolism and phagosome pathways between the control groups and AF simulated groups before treatment.

Specific AF-related gene variants were studied in rat and murine model systems and in one human iPSC atrial CM study [23,55,56,57,61]. These variants include SNPs, point mutations, and the introduction of transgenic constructs or gene knockouts and indicate the involvement of several signaling pathways in structural remodeling, electrical remodeling and AF susceptibility. Transcriptomic profiling in diabetic mouse models suggests a central role for MAPK10 signaling in enhanced AF susceptibility [56]. IPSC-derived human atrial cardiomyocytes with an AF-linked sodium ion channel mutation showed increased expression of NO signaling components, while subsequent inhibition of NO rescued the increase in late sodium current caused by the mutation [55]. Atrial expression signatures from TNF-α knock-out mice exhibit a reduced fibrotic response to intense exercise, which highlights the role of mechanosensing as a permissive factor in structural remodeling [57]. Treating rat neonatal cardio-fibroblasts with AF-related mutant NPA activates innate immunity pathways NLR, TNF-α and NFkB and facilitates structural remodeling through inflammatory leukocyte infiltration, increased fibrosis and differentiation of cardiac fibroblasts into myofibroblasts [61].

Overall, the general overlap in AF-related pathways resulting from RNA-seq studies from experimental AF model systems and AF patient groups is substantial and includes cardiac conduction, ECM remodeling, inflammation and cell–cell interaction. Inflammatory signaling and cellular response pathways are widely featured in experimental model studies of AF, whereas AF patient cohort studies seem to highlight more tissue level processes in ECM remodeling and regulatory signaling pathways. For example, the Wnt pathway is a common mechanism in patient studies, but does not surface in any of the experimental models. Both clinical and experimental studies use bulk RNA-seq or RNA sequencing of atrial CMs specifically. Cellular composition within the model systems is a critical consideration when translating findings to a human context. To illustrate, cardiomyocytes and fibroblasts are major cell types in adult human hearts, as either cell type represents between 20–30% of total cardiac cells, and immune cells only make up 10% of the total [66,67]. In contrast, the relative amounts of cardiomyocytes (~50%) and immune cells (~20%) both surpass the number of fibroblasts (~10%) in adult murine hearts [66].

As the experimental AF model groups involve artificially induced AF, their scope is limited to the translational response to atrial tachycardia and the resulting mechanisms for atrial remodeling. Notwithstanding, animal model AF research has inspired our current understanding of electropathology as a driver for AF, by introducing the concept of ‘AF begets AF’ [68,69]. Effects of aging on structural and electrical atrial remodeling and accompanying increased AF susceptibility have been demonstrated in mouse, rat and canine models [70,71,72,73]. Transcriptomic research on aging as an AF risk-factor might currently be overlooked, while Animal models could facilitate research on the transcriptional impact of aging on atrial remodeling and AF through RNA-seq studies.

## 4. Comparing Results of Patient and Large Animal RNA-Seq Studies

In the interest of providing an overview of AF-associated pathways and the overlap in differentially expressed genes within patient AF studies and large animal studies, DEGs found between SR and PeAF groups in the studies of Zeemering et al. and Van den Berg et al. are compared to DEGs found in the studies of LeBlanc et al. and Alvarez Franco et al. [9,10,64,65]. This data was available as Appendix A to the published articles. We chose to focus on the two recent studies that have sequenced the largest AF cohorts. The two large animal studies by LeBlanc et al. and Alvarez Franco et al. were chosen because of our interest in large experimental AF models as analogues for patient cohorts.

To provide an overview of common pathways associated with AF-associated genes, we mapped all differentially expressed genes to Kyoto Encyclopedia of Genes and Genomes (KEGG) pathways. Counts of DEGs mapped to specific pathways are displayed in Figure 1.

Interestingly, we found a substantial overlap in the AF-related pathways in patient and large animal studies. Three major categories of pathways surfaced in our comparison: metabolic pathways, various signaling pathways, and disease-related pathways. Cancer, infection and disease-related pathways are included in KEGG as broader pathways, that commonly include the PI3K-Akt signaling pathway, as well as MAPK signaling, cytokine-cytokine receptor interaction, focal adhesion, and calcium signaling pathways.

Despite the overlap in pathways identified in human and animal AF studies, the overlap in specific DEGs underlying these KEGG pathways was less prominent (Figure 2).

Figure 2A, indicates that 24% (54/221) of the differentially expressed genes found in the Zeemering et al. study, are also found in the Van den Berg et al. study [9,10]. When further overlaying the human studies with the two large animal studies, there is only a single shared gene: CKAP4. CKAP4 encodes a cytoskeleton-associated protein that enables RNA binding in myocardial tissues, binds with integrin, and is involved in PI3K-Akt signaling, MAPK signaling and mitochondrial function [74,75,76,77]. One might expect a larger overlap between the two human studies, and the lack of overlap between the top 100 most significant DEGs is especially remarkable (Figure 2B). The only DEG showing overlap in the two human studies presented is PPP1R12C, a myosin phosphatase subunit. The overlap between top 100 most significant DEGS found by Leblanc et al. and Alvarez Franco et al. consists of 2 extracellular matrix proteins and a cytoskeletal protein [64,65]. These specific structural proteins thereby form the consistent factor when comparing all four studies. It is important to note, that the comparison between the studies by Van den Berg et al. and Zeemering et al. is not flawless [9,10]. Zeemering et al. has matched the disease background of their SR and AF groups, whereas Van den Berg et al. has not [9,10]. As a result, the DEGs identified by Van den Berg may represent the transcriptional impact of ischemic heart disease or valve disease in atrial tissues over the impact of AF. Nevertheless, this example provides a clear illustration of the difficulties in comparing RNA-seq studies on patient cohorts.

The difference in number of DEGs found in the Zeemering and Van den Berg studies is striking, especially regarding Van den Berg et al. only included protein coding genes in their analysis [9,10]. Dissimilarities in their methods might explain this difference, and include the higher standard for RNA-integrity upheld by Van den Berg (RIN > 7.5 vs. RIN > 6.0) and the sequencing method. Where Van den Berg obtained 100 bp paired end reads from the NovaSeq s600, Zeemering obtained 75 bp paired end reads from the NovaSeq 500 [9,10]. Sequencing depth per sample is not mentioned in either paper. Variability of control and patient groups may also play a role in the high number of DEGs found by Van den Berg et al. Tissue samples from SR patients were excised during CABG or valve surgery, whereas tissue samples from both AF groups were excised during thoracoscopic ablation surgery. The ratio of patients suffering valve disease and/or coronary artery disease are not specified for any group, which hampers further assessment of variability between SR and AF patient groups.

## 5. Challenges and Perspectives in RNA Sequencing Technologies

Transcriptomics studies have underscored the complexity of AF, as it involves numerous molecular pathways (Table 1 and Table 2), remodeling at the tissue level, and electrical conduction disorders [10]. Changes in gene expression found through transcriptomic analysis do not directly reflect changes in, for example, ionic currents in atrial cardiomyocytes. As such, RNA-seq is best employed to highlight possible mechanisms and pathways that impact AF-related cell functions. The resulting molecular processes serve as targets for enabling further mechanistic research methods that cannot be applied transcriptome-wide. It is important to note that mechanisms and pathways highlighted by individual clinical RNA sequencing studies may be influenced by the compositions of their patient groups, especially in studies where SR and AF groups are not matched. Comprehensive analysis of multiple transcriptomics studies therefore has added value in identifying potential key mechanisms in AF occurrence, especially regarding atrial remodeling. The electrophysiology of AF is intrinsically bound to atrial remodeling, which serves as a ubiquitous factor in the otherwise heterogeneous AF. Current patient-cohort studies analyze RNA that is extracted from whole atrial tissue. Left and right atrial appendage tissues are appropriate for studying the electropathology of AF, as both structural remodeling and electrical conduction disorders can be observed in these tissues in the context of AF [1,78,79]. However, a drawback of sequencing bulk RNA from the atrial tissue is that the observed changes in gene expression mostly embody large scale shifts, instead of specific mechanisms in individual cell types. An alternative approach, the recently emerging technology of single-cell RNA sequencing (scRNA-seq) and single-nucleus sequencing (snRNA-seq) offer the potential to capture cell type heterogeneity at an unprecedented resolution. These sequencing approaches are able to detect changes within a tissue by profiling the transcriptomes of individual cells or nuclei [80]. Universal references for cell type labels, healthy cardiac cell type composition and expression profiles is readily accessible in the form of the human heart cell atlas [67]. Cell type-resolved differential expression analysis allows for precise identification of disease mechanisms in complex, heterogeneous tissues such as the atrial myocardium in AF. The first studies using snRNA-seq research in ischemic, hypertrophic and dilated cardiomyopathies, have identified disease-related shifts in cellular composition, common transcriptomic profiles between cardiomyopathies that indicate HF as clinical outcome, and have contributed to the identification of novel therapeutic targets for HF [81,82,83].

Another emerging technology that provides single-cell resolution is spatial transcriptomics. Spatial transcriptomics allows for the analysis of gene expression patterns in situ, by quantifying localized gene expression across tissue sections. This will leave the spatial context intact, which is otherwise destroyed by cell or nuclei isolation protocols. At this moment the resolution of spatial transcriptomics is substantially lower than single-cell RNA sequencing. Nevertheless, given the fast pace of technological developments, it is expected that single-cell spatial resolution in combination with genome scale gene expression will be available in the near future. It will be interesting to implement spatial transcriptomics to the atrial myocardium from patients suffering from AF. This will undoubtedly shed light on mechanisms of structural remodeling, inflammation and/or the epicardial contribution to AF.

Despite the advantages of spatial transcriptomics and single-cell sequencing, there are also challenges related to these new technologies. Technical variability and low sequencing coverage due to limited amount of starting material, variability of cell size and low cDNA conversion efficiency inevitably result in higher technical noise. As such, these technologies are not suitable at the moment to interrogate lowly expressed transcripts, and accurately annotate alternative splicing and isoform diversity. It is expected that these hurdles will be overcome in the near future, and that sequencing data will be integrated with other omics technologies, such as proteomics, metabolomics, and epigenomics. This will inevitably provide a more comprehensive understanding of the biological systems in the diseased heart.

## 6. Conclusions

Insights into AF-induced transcriptomic alterations in atrial tissue samples provide a deeper understanding of molecular mechanisms driving AF. RNA-seq facilitates transcriptomic analyses and discovery of differences in gene expression profiles between patient groups. Compared to controls, AF is associated with downregulation of mostly signaling pathways that guide a cellular response to an extracellular signal, including MAPK signaling and Wnt signaling, whereas upregulated pathways include intracellular signaling pathways (PI3KT-akt, Rap1), cellular response to external stimuli (MAPK), and cardiac muscle contraction. Structural and electrical remodeling mechanisms are also the focus of previous microarrays studies, but RNA-seq adds a layer of depth by including the analysis of ncRNAs as regulatory transcription elements [5]. Experimental model systems for AF identify pathways including cardiac conduction, ECM remodeling, inflammation and cell–cell interaction. When comparing large animal model and AF patient cohort studies, substantial overlap is found on the pathway level. However, overlap in specific DEGs is very limited, especially when comparing the most significant DEGs. With a look into the future, mechanistic AF research will entail single-cell resolution transcriptomics and multi-omics integration, enabling further identification of specific molecular targets with a prime role in the electropathology of AF.

## Figures and Tables

**Figure 1 cells-12-02242-f001:**
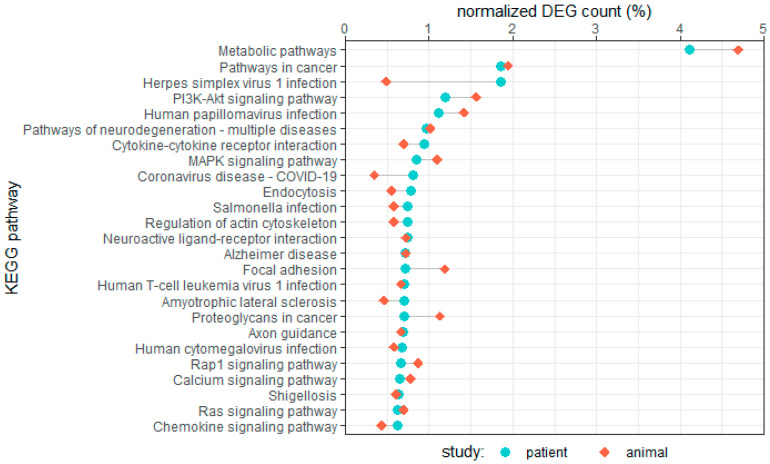
Enriched KEGG pathways in two AF patient studies and two AF animal studies. Counts of differentially expressed genes mapped to pathways were normalized over the total mapped differentially expressed genes for their respective study type (patient and animal). The top 25 pathways for human studies are displayed in this graph [9,10,64,65].

**Figure 2 cells-12-02242-f002:**
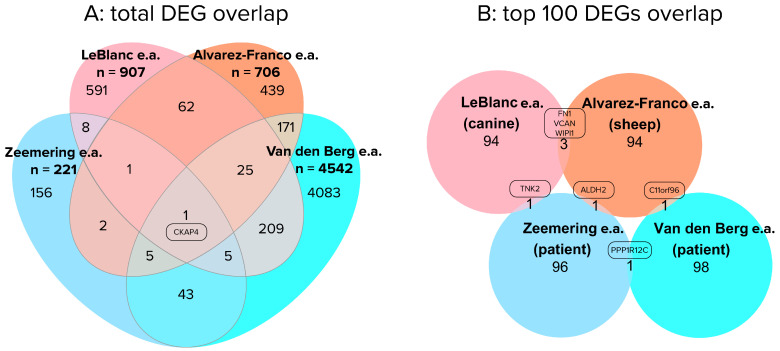
Overlap between AF-related differentially expressed genes found in two patient studies (SR versus PeAF) and two animal studies: (**A**) for total differentially expressed genes found, and (**B**) for top 100 most significant differentially expressed genes by adjusted *p*-value. Differentially expressed genes that did not have a (human ortholog) gene symbol assigned were omitted from the overlap analysis. Additionally, a gene list for total differentially expressed gene overlap is included in Appendix A [9,10,64,65].

**Table 1 cells-12-02242-t001:** Overview of RNA sequencing studies in human AF cohorts.

Source	Tissue	RNA	SR Patients		PAF Patients		PeAF Patients		Implicated Pathways
			Number	Surgical Indication	Number	Surgical Indication	Number	Surgical Indication	
[9]	LAA, RAA	mRNA, lncRNA	91	CABG, valve	53	CABG, valve	51	CABG, valve	inflammation, cellular respiration, fibrosis, cell division
[10]	LAA	mRNA	22	CABG, valve	22	thoracoscopic ablation	20	thoracoscopic ablation	EMT, ECM remodeling, angiogenesis
[11]	LAA, RAA	mRNA	5	CABG, valve	0	-	5	CABG, valve	wnt, circadian entrainment, ligand-gated ion channels, intracellular transport, ECM and tissue structure
[12]	LAA, RAA	mRNA, miRNA	1	healthy donor	0	-	3	Maze surgery	-
[13] *	LAA, RAA	mRNA, lncRNA	5	CABG, valve	0	-	5	CABG, valve	Adipose tissue accumulation, TGFB signaling, ion channel remodeling
[14]	LAA	mRNA, lncRNA	3	healthy donor	3	Maze surgery	3	Maze surgery	ECM interaction, Insulin resistance, Ras/MAPK signaling
[15]	LA, RA	mRNA	2	valve	0	-	3	valve	cell differentiation and development, cellular metabolism
[16]	LA, RA	mRNA	2	valve	0	-	3	valve	chloride ion channels, focal adhesion pathway
[17]	LAA, RAA	miRNA, circRNA	6	valve	0	-	9	valve	protein binding, nucleotide binding, nucleoside phosphate binding
[18]	LAA	circRNA	4	healthy donor	0	-	4	surgical ablation	protein modification, tissue development, calcium ion signaling
[19]	LAA	circRNA	6	healthy donor;	0	-	9	valve	DCM, HCM, regulating pluripotency of stem cells, Hippo signaling, TGFB signaling
[20]	LA	miRNA, circRNA	6	valve	5	valve	3	valve	wnt, ECM remodeling, Cell junction organization, cardiac conduction ***
[21]	RA	mRNA, miRNA	2	atrial septal repair, tricuspid vegetation excision	0	-	3	valve	cell–cell adhesion, TNF-α signaling, p53 signaling, EMT
**Source**	**Tissue**	**RNA**	**SR patients**				**AF patients ****		**Implicated pathways**
			**number**	**surgical indication**			**number**	**surgical indication**	
[22]	RA	mRNA, lncRNA, miRNA, circRNA	7	valve			7	valve	Rap1 signaling, MAPK signaling, oxidative phosphorylation, cardiac muscle contraction, tight junction
[23]	LA, RA	mRNA	7; 3	healthy donor; heart transplant			5	heart transplant	ion channels, transcriptional regulation, Notch signaling
[24]	LAA	mRNA, lncRNA	14; 38	healthy donor; CABG, valve			213	CABG, valve, surgical ablation	calcineurin signaling, sarcomere organization, wnt RhoGAP
**Source**	**Tissue**	**RNA**	**No PoAF patients**				**PoAF patients**		**Implicated pathways**
			**number**	**surgical indication**			**number**	**surgical indication**	
[25]	LA	mRNA, lncRNA	41	valve			21	valve	wnt, cGMP metabolism

* Study uses data from Thomas et al., 2019 [11]; ** number of PAF and PeAF not clearly specified; *** pathways for top differentially expressed genes mentioned in study according to Pubchem Reactome; PAF: paroxysmal AF, PeAF: persistent AF, and Po-AF: post-operative AF.

**Table 2 cells-12-02242-t002:** Overview of RNA-seq studies in experimental AF models.

Source	Model	Tissue	Model Groups	Control *n*=	AF *n*=	AF Induction or Simulation	Implicated Pathways
[55]	Human iPSCs	atrial CMs	control, SCN5a mutant, gene correction control	2	1	-	nitric oxide signaling, cardiac ion channels
[56]	Mouse	LA	WT, db/db	18	18	intracardial pacing	MAPK10 signaling, IL-17 signaling, TNF signaling, cAMP receptor signaling
[57]	Mouse	LAA, LV	sedentary/exercised, WT/TNF-KO	Unclear	-	-	ECM remodeling, fibrosis, intercellular communication
[23]	Mouse	RA/LA- CM nuclei	control, iNICD	6	6	-	NOTCH signaling, cardiac conduction
[58]	Rat	RA	control, CIH	15	15	Hypoxia	cell cycle, p53 signaling, IL-17 signaling, NLR signaling, cell adhesion
[59]	Rat	LA, venous blood	control, curcumin treatment	4	6	Ach-CaCl_2_	collagen synthesis, lipid metabolism, inflammation, angiogenesis
[60]	Rat	LA	control, AF control, AF colchicine treatment	4	7	Ach-CaCl_2_	phagosome, IL-17 signaling
[61]	Rat	neonatal CFs	wNPA treatment, mNPA treatment	2	2	mutant human NPA treatment	NLR signaling, TNF-α, NF-κB
[62]	Rabbit	unclear	control, paced	3	3	intracardial pacing	KCNJ2 expression
[63]	Rabbit	RA	control, paced	3	3	tachypacing	calcium signaling, gap-junction, focal adhesion
[64]	Canine	LA	control, paced	6	12	tachypacing, tachypacing + AVB	ECM structure, muscle structure development, striated muscle cell differentiation, glutamate signaling
[65]	Sheep	LAA/RAA	control, paced	3	6	tachypacing	calcineurin signaling, chromatin structure, cell adhesion, ion channels, TGF-β signaling, SLIT/ROBO

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
