# Peer review of "Dissecting the Molecular Mechanisms Driving Electropathology in Atrial Fibrillation: Deployment of RNA Sequencing and Transcriptomic Analyses"

_cells, 2023, doi:10.3390/cells12182242_

Round 1

Reviewer 1 Report

The efficacy to treat atrial fibrillation (AF) is still unsatisfactory, which might be caused by an insufficient knowledge about the molecular pathways leading to the electrical conduction disorders. This review provides an overview of several AF RNA-seq studies to identify molecular pathways by using atrial tissue samples from humans and experimental models. Authors describe identified molecular pathways and further compared the results from different models and discussed contrasting findings.

The authors present a very informative overview of several AF RNA-seq studies. However, the quality of the manuscript should be improved by some modifications.

The completeness of the tables should be proved. For example, studies form Tsai are not mentioned (Tsai et al. (2016) Int. J. Cardiol. 222)

A table including ncRNAs, which were identified in RA tissue samples and their targets and implicated pathways should be included.

Differences in animal models should be described in more detail, particularly regarding the animal species.

The result from Figure 2B should be discussed and explained.

The manuscript should be rearranged by adding some subheadings. 

To make the reading of the article easier, it would be helpful to include a list of all abbreviations used.

The conclusion should be extended by discussing the differences in patient and control groups.

A description or legend of the supplementary file is missing.

English grammar should be improved.

Author Response

The efficacy to treat atrial fibrillation (AF) is still unsatisfactory, which might be caused by an insufficient knowledge about the molecular pathways leading to the electrical conduction disorders. This review provides an overview of several AF RNA-seq studies to identify molecular pathways by using atrial tissue samples from humans and experimental models. Authors describe identified molecular pathways and further compared the results from different models and discussed contrasting findings.

We thank reviewer 1 for his/her insightful comments and suggestions.

The completeness of the tables should be proved. For example, studies form Tsai are not mentioned (Tsai et al. (2016) Int. J. Cardiol. 222)

The suggested paper by Tsai et al. describes a microarray study, whereas our overview in table 1 is exclusively committed to RNA-seq studies. In the revised version of the manuscript we included a short description (on page 2 and 12) to emphasize the difference and to mention previous studies performed with microarrays in the context of AF.

A table including ncRNAs, which were identified in RA tissue samples and their targets and implicated pathways should be included.

ncRNAs that were identified in the discussed RNA-seq studies were already discussed in the main text of paragraph, for example in lines 169-173 and 189-196 . We believe that adding a separate table for such a specific group of transcripts would disrupt the current structure and scope of chapter 2. Moreover, the targets or pathways of most ncRNAs are currently not known. In Table 1 we now added an extra column mentioning the type of RNA, including ncRNA detected.

Differences in animal models should be described in more detail, particularly regarding the animal species.

The differences in animal models that are essential to this review are the scope of the transcriptomic research for the different types of experimental models and, as well as the outcomes. These are described in chapter 3, line 244. We believe that further and more  detailed description of the different animal species that are used in AF research would distract from the current focus on the different types of experimental RNA-seq AF research and their outcomes. The available animal models for AF research in general has already been reviewed in detail by Schüttler et al.: (Schüttler, D., Bapat, A., Kääb, S., Lee, K., Tomsits, P., Clauss, S., & Hucker, W. J. (2020). Animal Models of Atrial Fibrillation. Circulation research, 127(1), 91–110. https://doi.org/10.1161/CIRCRESAHA.120.316366). We added this reference to the manuscript (Page 6)

The result from Figure 2B should be discussed and explained.

Thank you for this remark. To further explain the (lack of) overlap, found between the top 100 most significant DEGs, that is displayed in figure 2B, we have added a short discussion on the few DEGs that do overlap between the studies at line 338.

The only DEG that overlaps between the two human studies here is PPP1R12C, a myosin phosphatase subunit. The overlap between top 100 most significant DEGS found by Leblanc et al. and Alvarez Franco et al. consists of 2 extracellular matrix proteins and a cytoskeletal protein. Specific structural proteins thereby form the consistent factor when comparing these four studies.’

The manuscript should be rearranged by adding some subheadings.

To further emphasize the structure of the manuscript we have added subheadings to chapters 2 and 3 to divide them into more digestible sections.

To make the reading of the article easier, it would be helpful to include a list of all abbreviations used.

Good suggestion. We have now compiled a list of abbreviations and their respective definitions and have added this to the supplementary information.

The conclusion should be extended by discussing the differences in patient and control groups.

We have now added a sentence that describes the impact of differences in SR and AF groups to the paragraph that introduces the studies presented in Table 1 at line 88. This impact is further discussed in the concluding paragraph of chapter 5 at line 371.

‘It is important to note that mechanisms and pathways highlighted by individual clinical RNA-sequencing studies may be influenced by the compositions of their patient groups, especially in studies where SR and AF groups are not matched. Comprehensive analysis of multiple transcriptomics studies therefore has added value in identifying potential key mechanisms in AF occurrence.’

A description or legend of the supplementary file is missing.

We have added a short description of the supplementary file under supplementary files at line 435.

‘Table S1: List of overlapping differentially expressed genes, column names represent the studies included in the comparison. DEGs that overlap between these studies are mentioned by gene symbol in the rows below.

Table S2: List of abbreviations’

Comments on the Quality of English Language

English grammar should be improved.
We have looked into this, but without any specific suggestions or examples on where to improve the English grammar, only some minor modifications were made.

Reviewer 2 Report

The authors provide a well-written overview of transcriptome analysis through RNA sequencing of atrial tissue in the setting of atrial fibrillation.

I have a few remarks that should be taken into account in this manuscript:

-          The authors focus on transcriptome analysis through RNA sequencing in atrial fibrillation. Similar studies have been performed using microarray data. Thus far, there are more transcriptome data available on atrial fibrillation based on microarrays than on RNA sequencing. Therefore, the authors should discuss (shortly) these data, or, at least, mention that they exist and explain why they chose not to use them.

-          Table 1 should include information on what type of RNA was studied: mRNA, non-coding RNA, or both.

-          Figure 2: It is not clear to me whether the data represented in this figure contain DEGs between PAF and SR, PeF and SR, or both PAF and PeF vs SR.

-          The sentence on lines 299 and 300 should be rewritten, it looks like it is mixed up.

-          To my opinion, the results obtained by Van den Berg et al. and Zeemering et al. can not be compared since the SR group used by Van den Berg was not matched to the AF groups according to underlying pathology, whereas Zeemering used matched AF and SR groups. DEGs identified by Van den Berg may be mainly related to ischemic heart disease or valve disease.

-          References seem to be missing from the manuscript: Yang et al. Front Cell Dev Biol 2021; 9: 722671: They identified differential mRNAs and lncRNAs in atrial fibrillation using RNA sequencing. Wang et al. Cells 2022 Aug 24;11(17):2629: They identified differential mRNAs and miRNAs in atrial fibrillation using RNA sequencing.

Author Response

The authors provide a well-written overview of transcriptome analysis through RNA sequencing of atrial tissue in the setting of atrial fibrillation.

We appreciate the favorable and constructive comments of the reviewer.

-          The authors focus on transcriptome analysis through RNA sequencing in atrial fibrillation. Similar studies have been performed using microarray data. Thus far, there are more transcriptome data available on atrial fibrillation based on microarrays than on RNA sequencing. Therefore, the authors should discuss (shortly) these data, or, at least, mention that they exist and explain why they chose not to use them.

The reviewer raises a relevant point. We have now included a short paragraph in the introduction that mentions microarrays as an extensive source of transcriptomics data regarding AF research and refer the reader to the excellent review on coding transcriptomics by Steenman, published in 2020. Furthermore, we have added a sentence in the conclusions chapter at line 425 that recapitulates the similar scope of microarray studies to the reviewed RNA-seq studies and underlines the added benefit of analysing ncRNAs through RNA-seq.

‘Transcriptome analysis is a powerful tool for identifying molecular disease mechanisms. Extensive AF related gene expression data has been collected through atrial tissue and peripheral blood microarray studies. Microarray studies on AF and the insights they provide have been comprehensively reviewed by Steenman [5]. During the last decade, microarray studies have gradually been superseded by nucleotide sequencing technologies for transcriptome analysis due to their improved specificity to detect transcripts, and specifically isoforms.’

‘Structural and electrical remodeling mechanisms are also the focus of previous microarrays studies, but RNA-seq adds a layer of depth by including the analysis of ncRNAs as regulatory transcription elements (Steenman, 2020).’

-          Table 1 should include information on what type of RNA was studied: mRNA, non-coding RNA, or both.

Thank you for this suggestion, we have added a column to table 1 that specifies the mapping of mRNA, lncRNA, miRNA and/or circRNA by the studies included in the table.

-          Figure 2: It is not clear to me whether the data represented in this figure contain DEGs between PAF and SR, PeF and SR, or both PAF and PeF vs SR.

We thank the reviewer for pointing this out. We have now added a clarification in the main text of chapter 4 at line 302 and in the caption of figure 2 at line 326, stating that the overlap contains DEGs between SR and PeAF for both patient studies.

-          The sentence on lines 299 and 300 should be rewritten, it looks like it is mixed up.

Thanks for catching this mistake. The order of this sentence was indeed mixed up and you can find the rewritten sentence at line 331.

-          To my opinion, the results obtained by Van den Berg et al. and Zeemering et al. cannot be compared since the SR group used by Van den Berg was not matched to the AF groups according to underlying pathology, whereas Zeemering used matched AF and SR groups. DEGs identified by Van den Berg may be mainly related to ischemic heart disease or valve disease.
We concur with this opinion and have added a short discussion about the problem of comparing these two studies (See line 342). We maintain this inherently problematic comparison as an integral part of this review, as it provides a poignant illustration of the challenges in comparing outcomes of current RNA-sequencing research on AF patient cohorts.

‘It is important to note here that the comparison between the studies by Van den Berg et al. and Zeemering at al. is not flawless. Zeemering et al. has matched the disease background of their SR and AF groups, whereas Van den Berg et al. has not. As a result, the DEGs identified by Van den Berg may represent the transcriptional impact of ischemic heart disease or valve disease in atrial tissues over the impact of AF. Nevertheless, this example provides a clear illustration of the difficulties in comparing RNA-seq studies on patient cohorts.’

-          References seem to be missing from the manuscript: Yang et al. Front Cell Dev Biol 2021; 9: 722671: They identified differential mRNAs and lncRNAs in atrial fibrillation using RNA sequencing. Wang et al. Cells 2022 Aug 24;11(17):2629: They identified differential mRNAs and miRNAs in atrial fibrillation using RNA sequencing.

 We have included the study by Wang et al. and have added it to table 1 and shortly discuss the results at line 192. We did not include the study by Yang et al., as no details on their patient groups were available.

‘Interaction mapping between differentially expressed miRNAs and mRNAs in the RA of PeAF patients suggests suppression of SDC1 expression by miR302b-3p (Wang e.a., 2022). This negative interaction was validated in vitro and is reportedly involved in cardiac fibrosis through TGF-β/Smad2 signaling (Schellings e.a., 2010; Francogiannis, 2010).’

Reviewer 3 Report

The authors provide a succinct review of the current transcriptomic data in atrial fibrillation. The authors discuss how analysis of sequencing analyses may provide insight on molecular mechanisms in atrial fibrillation. Data from studies in human atrial fibrillation as well as animal models of atrial fibrillation is discussed.

The article is well-written. The table provides a helpful summary of the discussed studies. The figures complement the results discussed in the text. In addition to highlighting the interesting findings across the human and experimental studies, discussion of limitations of the studies such as read depth in sequencing analyses and lack of inclusion of comorbidities in characterizing patients is provided. My main comments are as follows:

1.     In the abstract, the authors discuss the promise of sequencing analysis to provide molecular insights on atrial fibrillation. However, as an example, a small change in gene expression could manifest as a large change in ionic currents, making it challenging to understand the impact of changes identified in sequencing results. Further discussion of the limitations of sequencing in atrial fibrillation should be provided.

2.     As there may be multiple mechanisms by which patients develop atrial fibrillation (e.g. genetic predilection vs inflammation especially in post-operative setting vs hemodynamic stress from valve disease), grouping AF patients for gene expression studies may be of limited yield. The authors recognize this limitation in part as some of the articles discussed do not distinguish which patients were undergoing CABG vs valve surgery, for example. But further discussion of the heterogeneity of atrial fibrillation should be discussed in the context of applying sequencing analyses.

3.     A comment on increased prevalence of atrial fibrillation among the elderly population, in the context of the aging population as mentioned, should be provided. Furthermore, discussion of age at time of tissue procurement should be provided. Is age taken into account in animal models discussed? This is of particular note in large animal studies in which the use of older animals may be cost prohibitive, further limiting translation of findings to humans.

4.     Additionally, patients with history of atrial fibrillation but in sinus rhythm at time of surgery were included in sequencing studies. This is not clear from the description of the human studies provided as these patients in sinus rhythm would be neither controls or post-operative atrial fibrillation patients.

5.     The ideal target tissue for atrial fibrillation studies should be discussed. Tissues analyzed in human studies are atrial appendage tissue, which may not be the ideal target tissue for atrial fibrillation given the importance of pulmonary veins. Furthermore, developmental processes involved in atrial fibrillation may not be adequately captured in studying older patients undergoing open heart surgery.

6.     The statement that authors of one study conclude “most transcriptional changes observed in PeAF are a consequence of AF, rather than a cause” highlights the issue that gene expression profiling may identify associations with atrial fibrillation but may be limited to identify the molecular underpinnings of AF. A more balanced discussion of the limitation of applying sequencing technologies in atrial fibrillation should be provided. Furthermore, in the discussion of Van den Berg et al (ref 9), the claim “[t]hese DEGs may represent genes that are involved in the development and progression” does not appeared supported as it is not clear that the findings are a result of rather than a cause of atrial fibrillation.

7.     Rather than list some of the studies, some more thoughtful discussion of the studies would be helpful. For example, the data of colchicine and curcurmin in a rat model of atrial fibrillation appears weak, with atrial fibrillation duration decreasing by only a few seconds, suggesting limited effectiveness, and limiting the utility of evaluating transcription changes with this treatment and translational potential of this research.

Author Response

The authors provide a succinct review of the current transcriptomic data in atrial fibrillation. The authors discuss how analysis of sequencing analyses may provide insight on molecular mechanisms in atrial fibrillation. Data from studies in human atrial fibrillation as well as animal models of atrial fibrillation is discussed.

The article is well-written. The table provides a helpful summary of the discussed studies. The figures complement the results discussed in the text. In addition to highlighting the interesting findings across the human and experimental studies, discussion of limitations of the studies such as read depth in sequencing analyses and lack of inclusion of comorbidities in characterizing patients is provided. My main comments are as follows:

We express our gratitude to reviewer 3 for sharing their detailed and constructive insights on the manuscript.

  1. In the abstract, the authors discuss the promise of sequencing analysis to provide molecular insights on atrial fibrillation. However, as an example, a small change in gene expression could manifest as a large change in ionic currents, making it challenging to understand the impact of changes identified in sequencing results. Further discussion of the limitations of sequencing in atrial fibrillation should be provided.

RNA-sequencing is used as a method to identify pathways that underlie disease pathophysiology and may contribute importantly to our understanding of the molecular disease mechanisms to support development of diagnostics and therapeutics, which is mentioned as the main goal in the introduction. Indeed, changes in gene expression do not directly reflect protein and ion channel function as post-transcriptional modifications are not taken into account. We emphasized this matter by adding to chapter 5 (line 366).

‘Changes in gene expression found through transcriptomic analysis do not directly reflect changes in, for example, ion currents in atrial cardiomyocytes. As such, RNA-seq is best employed to highlight possible mechanisms and pathways that impact AF-related cell functions. The resulting selection of molecular processes serve as targets for enabling further mechanistic research methods that cannot be applied transcriptome-wide.‘

  1. As there may be multiple mechanisms by which patients develop atrial fibrillation (e.g. genetic predilection vs inflammation especially in post-operative setting vs hemodynamic stress from valve disease), grouping AF patients for gene expression studies may be of limited yield. The authors recognize this limitation in part as some of the articles discussed do not distinguish which patients were undergoing CABG vs valve surgery, for example. But further discussion of the heterogeneity of atrial fibrillation should be discussed in the context of applying sequencing analyses.

We agree that multiple mechanisms may lead to AF, and therefore underlying molecular pathways may differ. RNA-seq data in patients with AF may reveal pathways that are altered due to the tachycardia itself, as discussed by the study of Van den Berg (line 342). We further emphasize this challenge in an additional sentence at line 371, and reinforced the value of these specific type of studies as cases for elucidating atrial remodeling, which is a central problem to AF research.

‘It is important to note that mechanisms and pathways highlighted by individual clinical RNA-sequencing studies may be influenced by the compositions of their patient groups, especially in studies where SR and AF groups are not matched. Comprehensive analysis of multiple transcriptomics studies therefore has added value in identifying potential key mechanisms in AF, especially regarding atrial structural and molecular remodeling. As electrophysiology of AF is intrinsically bound to atrial remodeling, it serves as a ubiquitous factor in the otherwise heterogeneous AF.’

  1. A comment on increased prevalence of atrial fibrillation among the elderly population, in the context of the aging population as mentioned, should be provided. Furthermore, discussion of age at time of tissue procurement should be provided. Is age taken into account in animal models discussed? This is of particular note in large animal studies in which the use of older animals may be cost prohibitive, further limiting translation of findings to humans.

The increased prevalence of AF as an age-related disease is mentioned in the introduction, where AF is described as “the most common age- related arrhythmia” (line 33) and later on “As the life expectancy of the worldwide population is increasing, the steep rise in in incidence of AF in the general population is becoming an urgent public health issue” (line 36).

Age at time of tissue procurement is not discussed in the context of patient cohort studies, as this factor is generally matched between SR and AF groups. Difference in age does come into consideration when healthy donor tissue is used as the control and we have added short discussion on this to line 88: ‘Separation of AF-related expression patterns is further complicated in studies that have not matched their SR and AF groups, where differential gene expression may reflect underlying cardiac disease over AF. Other factors, like aging, will provide additional background signal when healthy donor tissue is used as the control.’

Aging animal models are a very interesting consideration. The animal model studies involve artificially induced AF, which limits their scope to the atrial transcriptional response to atrial tachycardia. The effect of aging on this response is a fascinating research prospect, but has not been studied in animal models using RNA-seq. We have added an additional concluding paragraph to chapter 3 that discusses this prospect.

Line 290: ‘As the experimental AF model groups involve artificially induced AF, their scope is limited to the translational response to atrial tachycardia and the resulting mechanisms for atrial remodeling. Notwithstanding, animal model AF research has inspired our current understanding of electropathology as a driver for AF, by introducing the concept of ‘AF begets AF ‘ [67, 68]. Effects of aging on structural and electrical atrial remodeling and accompanying increased AF susceptibility have been demonstrated in mouse, rat and canine models [69-72]. Transcriptomic research on aging as an AF risk-factor might currently be overlooked, while animal models could facilitate research on the transcriptional impact of aging on atrial remodeling and AF through RNA-seq studies.’

  1. Additionally, patients with history of atrial fibrillation but in sinus rhythm at time of surgery were included in sequencing studies. This is not clear from the description of the human studies provided as these patients in sinus rhythm would be neither controls or post-operative atrial fibrillation patients.

Patients that have a recorded history of atrial fibrillation, but were in SR rhythm at the time of surgery are either specified as PAF patients or AF patients, depending on their specification in the respective studies. Patients with a known history of AF are not included in the control groups. The AF classification does not indicate rhythm at time of surgery, which is now specified at line 76.

  1. The ideal target tissue for atrial fibrillation studies should be discussed. Tissues analyzed in human studies are atrial appendage tissue, which may not be the ideal target tissue for atrial fibrillation given the importance of pulmonary veins. Furthermore, developmental processes involved in atrial fibrillation may not be adequately captured in studying older patients undergoing open heart surgery.

The relevance of discussing an “ideal target tissue” is debatable, when there is no current realistic alternative to obtain the tissues and models discussed in the review. As shown by the research group of De Groot, electrical conduction disorders are observed in both atria, including right and left atrial appendages and Bachmans bundle (Brundel, B.J.J.M., Ai, X., Hills, M.T. et al. Atrial fibrillation. Nat Rev Dis Primers 8, 21 (2022). https://doi.org/10.1038/s41572-022-00347-9). As atrial (appendage) tissue is subject to structural remodeling as well as electrical conduction disorders related to atrial fibrillation, it therefore can be utilized to study the molecular pathways that relate to electropathology.

We mentioned this item in the manuscript chapter 5, line 378:

‘Left and right atrial appendage tissues are appropriate for studying the electropathology of AF, as both structural remodeling and electrical conduction disorders can be observed in these tissues in the context of AF [1, 78, 79].’

  1. The statement that authors of one study conclude “most transcriptional changes observed in PeAF are a consequence of AF, rather than a cause” highlights the issue that gene expression profiling may identify associations with atrial fibrillation but may be limited to identify the molecular underpinnings of AF. A more balanced discussion of the limitation of applying sequencing technologies in atrial fibrillation should be provided. Furthermore, in the discussion of Van den Berg et al (ref 9), the claim “[t]hese DEGs may represent genes that are involved in the development and progression” does not appeared supported as it is not clear that the findings are a result of rather than a cause of atrial fibrillation.

A more balanced discussion of the limitations was added to chapter 5 as a response to comment number 2 (line 371).

Indeed the claim mentioned above does not account for whether the findings represent a cause or result of atrial fibrillation. We have now limited the claim to stating that the findings may represent genes involved in the progression of AF (line 126), as they are differentially expressed between progressive states of AF. Invoking the concept of AF begets AF, mechanisms that are the result of AF can represent the cause of AF progression.

  1. Rather than list some of the studies, some more thoughtful discussion of the studies would be helpful. For example, the data of colchicine and curcurmin in a rat model of atrial fibrillation appears weak, with atrial fibrillation duration decreasing by only a few seconds, suggesting limited effectiveness, and limiting the utility of evaluating transcription changes with this treatment and translational potential of this research.

Our primary aim is to report on the highlighted pathways as presented by the authors. I agree that the conclusions on the two animal intervention studies specifically could be worded more carefully, but we believe that a deliberation on the efficacy of the treatments is outside the scope of this review. We have rewritten the paragraph to emphasize our prioritization of the findings from comparing the control groups and AF simulated group over the treatment effects.

‘Treatment of mouse AF models with curcumin or colchicine, which are considered to protect against AF, affect gene expression in pathways associated with inflammatory regulation and fibrosis, and IL-17 signaling and renin secretion pathways respectively [58, 59]. The findings indicate a possible role for these pathways in experimental AF. More relevant is the differential expression that was found in inflammatory signaling, fibrosis, lipid metabolism and phagosome pathways between the control groups and AF simulated groups before treatment.’

Round 2

Reviewer 3 Report

The authors have satisfactorily addressed my concerns.